# Spatial Efficiency and Socioeconomic Efficiency in Urban Land Policy and Value Capturing: Two Sides of the Same Coin?

Jean-Marie Halleux [1,*], Berit Irene Nordahl [2] and Małgorzata Barbara Havel [3]

1 SPHERES Research Unit, Department of Geography, ECOGEO–Lepur, University of Liège, B11, Clos Mercator 3, 4000 Liège, Belgium
2 Institute for Urban and Regional Research, Oslo Metropolitan University, Postbox 4 St. Olavs Plass, 0130 Oslo, Norway
3 Department of Spatial Planning and Environmental Science, Faculty of Geodesy and Cartography, Warsaw University of Technology, 00-661 Warsaw, Poland
* Correspondence: jean-marie.halleux@uliege.be

**Abstract:** Land policies are commonly used to contribute to the implementation of the public policy of land-use planning (or spatial planning). In this sense, a spatially efficient land policy must allow the planning systems to achieve the land uses promoted by strategic planning. In addition to their role in land-use planning, land policies also contribute to public finance policies. From this perspective, a socioeconomically efficient land policy must allow public authorities to capture land value. The research presented in this article aimed to contribute to planning theory by initiating a reflection on the interactions between spatial and socioeconomic efficiency in land policy. In our view, to consider those two dimensions in a more integrated way can help in the development of the growing research field on land value capture. Our research is based on the general assumption that there are processes of mutual strengthening and blockages between spatial efficiency and socioeconomic efficiency in land policy. In order to test this assumption, an international comparison methodology was developed. In order to develop a robust methodological approach, our exploratory comparative approach is based on a theoretical framework that depicts an ideal planning system. This ideal planning system serves as a benchmark for two empirical case studies on Norway and Belgium. Through our research, we find evidence of the interdependency of spatial efficiency and socioeconomic efficiency. The results of the two case studies therefore indicate that our initial assumption is generally confirmed. However further investigations are needed to deepen our exploratory discussion of the topic.

**Keywords:** land policy; value capturing; spatial efficiency; socioeconomic efficiency; Norway; Belgium

## 1. Introduction

Land policy can be broadly defined as all actions implemented by public authorities to address the implications of the distribution of property rights concerning land [1,2]. Land policies are therefore pursued to achieve plural ambitions at the heart of legal, social, economic and environmental life [3]. The actions implemented within the framework of land policies can exhibit varying degrees of sophistication. They can also be more or less passive (for instance through the granting of a development permit) or interventionist and active (for instance, through the nationalisation of land or through the implementation of a land register). A fundamental and basic ambition of land policies is to ensure the effective operation of land and property markets (for instance through the proper functioning of the created land register). Apart from this basic ambition, land policies are essentially related to spatial ambitions and to socioeconomic ambitions [4]. The development of the exploratory research presented in this paper precisely aims to shed light on this duality between spatial and socioeconomic ambitions in urban land policy.

Land policies are commonly used to implement land-use planning (or spatial planning) [5,6]. In this sense, a spatially efficient land policy must allow spatial planners to

achieve the land uses promoted by strategic planning. In addition to their key role in land-use planning, land policies also contribute to public finance policies. From this perspective, a socioeconomically efficient land policy must allow public authorities to capture land value. As notably put forward by the OECD [7], land value capture has never been as important to the future of public authorities as it is today, especially for local authorities. Indeed, this funding source can be greatly helpful for locally meeting global challenges such as rapid urbanisation, deteriorating infrastructure, or climate change.

The starting motivation of the exploratory research presented in this article was to contribute to planning theory and to land policy theory by initiating a reflection on the interactions between the spatial and the socioeconomic ambitions in urban land policy. We consider that this theme has not yet been adequately addressed by scholars. To date, although they can be considered two sides of the same coin, spatial ambitions and socioeconomic ambitions tend to be understood in a separate way by academic researchers on land policy. In our view, to consider those two dimensions in a more integrated way can notably help in the development of the growing research field dedicated to land value capture [8].

Our investigations were developed following an initial assumption that there are processes of mutual strengthening and blockages between spatial efficiency and socioeconomic efficiency in urban land policy. The issue can therefore be analysed through the following four hypotheses:

**Hypothesis H1.** *In urban land policy, increasing spatial efficiency tends to increase socioeconomic efficiency.*

**Hypothesis H2.** *In urban land policy, increasing socioeconomic efficiency tends to increase spatial efficiency.*

**Hypothesis H3.** *In urban land policy, the inability to increase spatial efficiency tends to limit socioeconomic efficiency.*

**Hypothesis H4.** *In urban land policy, the inability to increase socioeconomic efficiency tends to limit spatial efficiency.*

In order to develop the issue and test the proposed four hypotheses, our methodology was based on a comparative analysis of planning systems. A planning system can be defined as "*the ensemble of institutions that are used to mediate competition over the use of land and property, to allocate rights of development, to regulate change, and to promote preferred spatial and urban form*" [9] (p. viii). Comparing planning systems is a classical and well-founded approach in planning research [10]. It refers to the modes of comparative learning that allows for learning from the past and from other places [11].

While the classic comparative analysis of planning systems considers real planning systems, the specificity of our approach lies in the consideration of a theoretical planning system. Considering a theoretical planning system as a reference model has helped in the development of a robust methodological approach. The conceptualised planning system depicts an ideal world where land policies efficiently contribute to the achievement of the land uses spatial planners seek and, simultaneously, efficiently contribute to redistributing land value to the community. In our research, we compared this ideal planning systems with two real planning systems through empirical case studies of Norway and Belgium. Norway and Belgium are both European countries. Norway belongs to the group of Nordic (Scandinavian) countries, and Belgium is located in northwest Europe, between France, The Netherlands, and Germany.

The rest of the article is organised into five sections. In Section 2, we present a theoretical framework with respect to spatial efficiency and socioeconomic efficiency in urban land policy. It is on the basis of this framework that we depict an ideal planning system that serves as a benchmark for the two empirical case studies of Norway and Belgium. In Section 3, we discuss the comparative methodology and the selection of the

two empirical case studies. In Section 4, we develop the two case studies, and the results are then discussed in Section 5. In Section 6, we conclude by emphasising that a focus on the interactions between the spatial and the socioeconomic ambitions in land policy should be more systematically considered in the growing research field on land value capture.

## 2. Theoretical Framework: Conceptualisation of an Ideal Planning System with Respect to Spatial Efficiency and Socioeconomic Efficiency in Urban Land Policy

This section presents the theoretical framework that we used for the two empirical case studies on Norway and Belgium. This theoretical framework is based on a literature review focused on the two concepts of spatial efficiency and socioeconomic efficiency. The more general concept of efficiency can have different meanings depending on the disciplines. Even within the discipline of planning, the concept of efficiency is multifaceted. To develop an in-depth analysis on this general concept is therefore beyond the scope of this paper (see [12] for a recent theoretical reflexion on the concept of efficiency in planning literature). Even though, our approach requires giving clear definitions of what we intend by the two more specific concepts of spatial efficiency and socioeconomic efficiency and it is from this perspective that we are now going to depict an ideal planning system with respect to those two aspects of spatial efficiency on the one hand and socioeconomic efficiency on the other hand.

### 2.1. Land-Use Planning and Spatial Efficiency in Urban Land Policy

Through their policy of land-use planning (or spatial planning), public authorities envision and determine the physical organisation of space and the spatial distribution of people and activities. To achieve these objectives, public authorities rely on the institutions which constitute their planning systems.

In our proposed ideal world, the public policy of land-use planning starts by participatory and cooperative processes aiming to define collective ambitions related to preferred spatial and urban form. These processes should be the first step in the development of strategic plans aiming to define, but also to communicate, those collectively defined spatial ambitions. In essence, strategic plans aim to reach agreements to coordinate a multitude of actors wishing to develop a multitude of projects through territories [13]. Strategic planning is therefore a complex undertaking, as various and potentially conflicting interests have to be considered, compared, and weighed. Moreover, development patterns must integrate various dimensions that relate to environmental goals (e.g., to reduce $CO_2$ emissions, to limit land take, to protect biodiversity, to limit the impact of floods) as well as to socioeconomic goals (e.g., to limit socio-spatial segregations, to attract new investments, to supply land for industries, etc.).

The success of a planning system relies on its ability to formulate coordinated spatial ambitions in strategic plans but also, and perhaps above all, to operationally implement the desired pattern of development through operational planning. Compared with the future-oriented strategic part of spatial planning, its operational part is more present-oriented. It focuses on the concrete realization of spatial ambitions defined by strategic planning and this is actually where land policy comes in. As put forward by J.-D. Gerber et al. [6], land policy focuses on operational planning and on the actual implementation of spatial ambitions defined by strategic planning. In this sense, the spatial efficiency in urban land policy is determined by its capacity to help the planning system to implement the spatial ambitions defined at the strategic level.

In the operational part of planning, two important land policy instruments play successively central roles in the development processes: the zoning plan and the granting of development permits. In an ideal planning system, planning authorities design zoning plans to specify the development modalities that make it possible to meet the ambitions of the strategic plans. Subsequently, planning authorities use the zoning specifications to evaluate development projects through approval procedures. If the projects are desirable in terms of the spatial qualities envisioned by the strategic plans, the development permits

should be granted. Conversely, they should be dismissed if they are in opposition to the specifications of the zoning plans and, consequently, in opposition to the related collective ambitions defined by strategic plans.

In the operational part of planning, an ideal planning system could rely on proactive land policy and/or on passive land policy. Proactive land policies in operational planning are initiated by public bodies, when they directly invest in land to achieve the land uses they seek. In contrast, under passive land policies, private bodies take responsibility for developments, and planning authorities enable development initiatives unless they are undesirable (see [14] for more developments on the differentiation between proactive and passive approaches in spatial planning).

The above description of the way planning systems should ideally operate is of course seldom found in the real world. In the real world, defining spatial ambitions which may receive broad collective support is very challenging. Moreover, even in the scenario of broad collective support, various factors can significantly limit the ability of planners to achieve the land uses they seek [15]. A major reason can be that there are scant political and financial resources available for planning systems. As a consequence, planners are not able to rein in other public decision makers [13]. Moreover, in the real world, active operational planning remains the exception and the strength of individual property rights often creates a serious gap between strategic spatial ambitions and what is actually made possible through passive operational planning [16].

### 2.2. Public Finance and Socioeconomic Efficiency in Urban Land Policy

2.2.1. Land Value Capture vs. Land Value Compensation

In addition to their crucial contribution in land-use planning, land policies must also contribute to public finance policies through land value capture. The socioeconomic ambition to redistribute land value to the community is politically debated [17], but its logical and philosophical justification relies on two fundamentals. The first fundamental is that an immovable that is built is actually a combination of two different components: The first component is the parcel of land, and the second component is the construction that rests on it. The second fundamental is that the market value of the parcel of land arises from collective actions and public investments more than from investments by the property owners [18]. In essence, land value capturing is therefore justified by the existence of what J.S. Mill [19] described as the "unearned increment", i.e., by "*an increase in the value of property through no work or no expenditure by the property owner*" [20] (p. 258).

Land value capture can take recurring forms as well as non-recurring forms [21]. Recurring forms, either annual taxes or taxes in case of sale or purchase, have typically more to do with the fiscal systems than with the planning systems. They therefore fall outside the scope of the present article. In contrast, non-recurring forms are typically embedded into planning systems, in particular through developer obligations. Developer obligations are a type of value capture instrument that can be defined as: "*contributions of property developers and landowners made in exchange for public bodies making decisions on land-use regulations that increase the economic value of their land and buildings*" [22] (p. 1). In particular, developer obligations can be used to extract contributions when decisions are being made regarding the development process, be they decisions about zoning plans or about development permits.

A comprehensive view on the issue of land value requires considering that collective actions, including decisions on planning regulations, may increase but may also decrease land value [23,24]. Both aspects of increasing and decreasing land value must therefore be considered regarding the socioeconomic efficiency in land policy. The first aspect, the upward effect, relates to the issue of land value capture: Does the right of property include *the right to the added value* created by collective actions and, in particular, by planning regulation and public works? The second aspect, the downward effect, *concerns compensation right*: does the right of property include the right to be compensated for any value decline



due to collective actions and, in particular, to value decline due to decisions on land use regulation?

When it relates to the development of an ideal planning system, we consider that socioeconomic efficiency in land policy depends on how land value increases and decreases are shared between private and public bodies [18]. In the case of increases, the larger the proportion of land value is captured by public authorities, the more efficient is the contribution of land policies to public finance policies [23]. Symmetrically, in the case of decreases, an ideally equitable planning system should integrate mechanisms of full compensation [2]. Indeed, intellectually, both aspects of the upward and downward effects concern the role that property rights on land play in a society. As a consequence, they should be considered in a symmetric way. Again, this actually relates to an ideal world. In the real world, most countries have decorrelated them, and the two aspects are not symmetric, either in political approaches or in law. Also within this domain, "*real-life laws and policies do not operate according to the axioms of elegant logic*" [25] (p. 762).

Concerning the relationships between land value capture and land value compensation, a specific issue relates to a situation of successive changes in land-use regulations. This can happen as successive changes in spatial ambitions can lead to successive changes in planning options and, therefore, to successive changes in land-use regulations. For example, this may be the case if, due to new collective ambitions for instance in urban containment or flood protection, greenfield building land must be rezoned back to non-constructible land. In such case, it makes sense to consider that (full) compensation would be legitimate and equitable only if the initial added land value had been the subject of a (full) public value capture. More generally, in the case of successive changes in land-use regulations, an ideal planning system should be able to successively apply mechanisms of both full value capture and full value compensation [24]. Again, everything indicates that in the real world, due to the decorrelation between the upward and downward effects, most planning systems are not able to proceed that way.

### 2.2.2. The Three Main Types of Value Capture Instruments

Following Alterman from 2012, three main types of value capture instruments can be differentiated: macro, direct and indirect. For Alterman, macro value capture instruments are embedded in overarching property right and land policy regimes motivated by a broad rationale and ideology. An extreme example of a macro value capture instrument is the nationalisation of land under communist regimes. Macro value capture instruments can also be applied in capitalist regimes, particularly as part of proactive land policies in operational planning. This can be illustrated by the well-documented "public land development" model that was widely applied in the Netherlands after the Second World War [26,27]. In this model, a public land developer buys all the land to be developed on the non-building land market; it readjusts the parcels into forms suitable for the desired development; and it sells those parcels to potential property developers on the building land market, be they private companies, housing corporations, or individual households.

In theory, the public land development model is ideal for greenfield developments in making it possible to simultaneously meet spatial ambitions (e.g., urban containment) and capture the added land value derived from the urbanisation process (the urban land rent). However, in practice, various factors suggest that the Dutch experience with public land development is difficult and even dangerous to transfer to other countries [14]. Indeed, it puts local authorities at high financial risk [28], and it also creates a "double hat" threat as the public developer, for instance a municipality agency, might operate too much out of motivation for financial aspects and too little on grounds of good land-use planning and spatial quality [29]. The double hat issue points out potential contradictions between spatial efficiency and socioeconomic efficiency in urban land policy. As a consequence, it goes against our initial—and idealistic—assumption of mutual strengthening between the two analysed dimensions. More precisely, it goes against Hypothesis 2 that in urban land policy, increasing socioeconomic efficiency tends to increase spatial efficiency.

Just as with macro value capture instruments, the ideological dimension is also of great resonance with direct instruments. Direct instruments are based on the principle of equity and they "*seek to capture all or some of the value rise in real property under the explicit rationale that it is a legal or moral obligation for landowners to contribute a share of their community-derived wealth to the public pocket*" [25] (p. 765). In contrast, the rationale behind indirect instruments is based on a more practical and pragmatic approach: "*indirect instruments do not seek to capture the added value for its own sake, simply because it is 'unearned', but in order to generate revenues (or in-kind substitutes) for specific public services*" [25] (p. 766). In practice, because they do not challenge fundamental ideological principles, indirect instruments are gaining prominence over the direct instruments. Among the various proliferating indirect value capture instruments, one finds various forms of developer obligations.

The typology developed by R. Alterman allows for understanding how the cultural conception of property rights over land impacts the allocation of resources. This probably explains why this typology has become a central reference in the growing literature on land value capture [24]. However, Alterman's typology also presents limitations, as land value capture instruments, including developer's obligations, are often based on a mixture of rationales [22] (p. 10). Moreover, as the typology is based on the justification and intention behind the value capture instruments, it says nothing about socioeconomic efficiency. In particular, it does not say that, compared with direct instruments that rely on a pure equity perspective, indirect instruments that rely on a pragmatic perspective are condemned to redistribute less value to the community. In fact, given the current state of knowledge, we can postulate that the three types of instruments could allow for redistributing all land value to the community. In theory, an ideal planning system could therefore be based on any or several of the three types.

### 3. Methodology: A Comparative Analysis of Theoretical and Real Planning Systems

The theoretical framework developed in the previous section led us to conceptualise a theoretical planning system that portrays an ideal world where land policies efficiently contribute to the achievement of the land uses spatial planners seek and, simultaneously, efficiently contribute to redistributing land value to the community. In our methodological approach, this ideal and theoretical planning system serves as a benchmark for studying real planning systems with respect to spatial efficiency and socioeconomic efficiency in urban land policy. In fact, we consider that any real planning system can potentially be compared with the "ideal" planning system we have just described. In the exploratory research presented in this article, we chose to work with Norway and Belgium.

Our decision to select Norway and Belgium derives from the fact that we are familiar with both planning systems as well as from the conjunction of resemblances and dissemblances between them. In both countries, as opposed to a country like the Netherlands [26,27], one finds limited proactive public land policies and, following the terms of Lind [30], a market-oriented land-use planning. Concomitantly, both countries are characterised by a great respect for individual property rights. In the two countries, it is typically considered that public authorities should not capture the unearned increment in land value. For various reasons, most Norwegian and Belgian stakeholders and policy makers do not recognise the analytical standpoint that land value (and how it changes) is to a very large extent solely the result of government decisions and/or collective actions.

Another resemblance between the two planning systems is the spatial ambition to limit urban sprawl. In the two countries, one finds strategic plans that insist on the need for land-use planning to shift the focus from greenfield development to urban regeneration. This resemblance relates to the resource orientation of planning [6]. This evolution was inspired by sustainability discourses and by the observation that greenfield development always becomes more problematic because of the related high consumption of resources. Since the 1990s, the discourse on sustainable development and limiting sprawl has led to the paradigm of the compact city, which has now become a hegemonic response to the challenges of sustainable development [31].

The compact city model entails a set of planning principles that aim to limit suburban sprawl and car dependency through compactness and density. In parallel, the model of the compact city has put forward the greening of the city to contribute positively to the quality of life in rather densely built environments. As will be illustrated by the Norwegian situation, the actual implementation of the compact city core principles requires substantial investments. Through land value capturing, this is where socioeconomic efficiency in urban land policy becomes vital.

However, in addition to substantial similarities, the two analysed countries also show striking differences when it relates to the actual implementation of the spatial ambition of urban sprawl limitation. In Norway, spatial planning has been able to limit the consumption of land resources and to favour an evolution towards the model of the compact city [32]. In contrast, in Belgium, spatial planning has failed to curb urban sprawl or, more generally, to implement spatial ambitions and concretely influence land uses [33]. As will be developed in the coming section, these differences in the concrete implementation of spatial ambitions have strong implications for the capacity of governments to redistribute land value to communities.

## 4. Empirical Case Studies

### 4.1. Norway: Socioeconomic Efficiency through the Need to Finance Ambitions for a High-Quality Compact City

Our case study of Norway shows that Norwegian land policies tend to combine spatial efficiency and socioeconomic efficiency rather successfully. This subsection presents this case study through an analysis of the keys to the success of Norwegian planners in the implementation of the compact city policy. The topic of the compact city was chosen because the performance of the Norwegian planning system against the spatial ambition of sprawl limitation has strong relationships with the socioeconomic performance of the Norwegian land policy against the socioeconomic ambition to contribute to public finance through land value capture. As will be discussed, this situation is consistent with Hypothesis 1 that, in urban land policy, increasing spatial efficiency tends to increase socioeconomic efficiency.

We start this subsection with a brief description of the Norwegian planning system. Subsequently, we consider how the compact city agenda is financially supported at the national level. This is followed by a focus on the limited application of the compensation right and then by another focus on development agreements. Conditions in the zoning specification and the related development agreements (*utbyggingsavtale* in Norwegian) are currently the main land value capturing instruments used by Norwegian municipalities to secure funding at the neighbourhood scale. Development agreements tie developer contributions to the granting of planning approvals, and this instrument therefore falls into the category of developer obligations.

### 4.1.1. General Organisation of the Norwegian Planning System

The ability of the Norwegian planning system to implement the spatial ambition of the compact city is inseparable from a hierarchical structure based on three levels: the national level, the regional level at the scale of the counties, and the local level at the scale of the municipalities. Less sprawl and more compact cities are high up on the national planning agenda in Norway. This is stated in the national government's expectation regarding all planning activities, as well as in the national policy for land use, transport, and housing [34–36]. The background departs from an almost thirty-year focus on transport-oriented planning [32,37], renewed and reinforced through the 2015 UN Paris agreement and Norway's global commitment to reduce $CO_2$ emissions.

In Norway, the compact city policy is not without local resistance, and the activity of Norwegian planners aiming to limit sprawl is not exactly a long calm river. Indeed, almost 8 out of 10 Norwegian homes are detached houses, and the policy of turning away from extension at the outskirts to population concentrations meets resistance not only from aggrieved landowners in the peripheries but also from proponents of more

traditional development ideas. However, as a consequence of the national policy, most regional masterplans now take up these growth principles, and the policy trickles down the planning tiers: from county masterplans to the municipal masterplans and local zoning plans, and hence to the granting of development permits [38].

In Norway, the national and the regional levels are active in strategic planning, and those authority levels count on municipalities to achieve operational planning. The local level is also active in strategic planning as the Norwegian municipalities are in charge of the drawing up of municipal masterplans (*kommuneplan* in Norwegian). This document is a vision document that includes land use options that are established in accordance with county masterplans. At the local level, one also finds two regulatory zoning plans: the local plan (*områdeplan* in Norwegian) and the detailed zoning plan (*reguleringsplan* in Norwegian) [39]. The granting of development permits is based on these instruments, which are therefore central in passive operational planning. Local plans describe certain areas in detail that notably relate to restrictions on the percentage of dwellings allowed or to the type of constructions. Detailed zoning plans are the most detailed land-use plans. They describe all the rules and restrictions that must be met when developing the area. These specifications may concern in particular the architecture of the buildings in terms of height, aesthetics, etc. As it will be developed below, development agreements are negotiated with the developers as part of the preparation of the detailed zoning plans.

### 4.1.2. Financing Ambitions for a High-Quality Compact City by the National Level: Funding-Against-Planning and Urban Growth Agreements

Compact growth policy and the limitation of car dependency are not only a question of developing public transport at the scale of the urban region: they are also a question of upgrading the liveability and attractiveness of the urban neighbourhood. This requires improvements of the public transport accessibility of the neighbourhoods, as well as greater quality of urban design and better access to open green areas [40]. At the urban region scale, the needed investments in spatial quality involve public transport efficiency, through high-quality and high-frequency metro and tram lines. At the neighbourhood scale, the required investments involve high-quality station areas with secure bike parking facilities, shops and services. The required investments also involve the upgrading of the surrounding areas with facilities such as new squares, parks, crosscutting green corridors, new pedestrian lines, bike and bus lanes, and the conversion of roads to urban streets.

When it comes to the Norwegian funding schemes of the qualitative compact city, we see a division between investments in public transport and investments in urban (neighbourhood) upgrading. The former is the responsibility of the state, through county administrations, whereas the latter is the responsibility of the municipalities. The success of the compact growth policy relies heavily on the system's ability to secure funding at both governance levels.

To leverage the compact city growth model, the Norwegian national government has introduced the urban growth agreement (UGA), a new and strong financial incentive tool of funding-against-planning. The UGAs are integrated in the national transport investment plan [41,42]. The first-generation UGA was introduced in 2014 and a second generation in 2019 [43]. UGAs are agreements between the national transport authority and the municipalities of the major metropolitan areas of the country [44]. Municipalities are the prime executive authority in spatial planning. When signing the UGA, the local authority must accept the need to plan for densification and redevelopment instead of sprawl. In return, they are awarded substantial investments in local public transport.

The core of the agreements is as follows: the state provides regional transport authorities with budgetary uplift to secure investments in public transport within the municipalities over a period of five to ten years. To be eligible, the municipalities must plan for compact growth, with the perspective of a society where more residents use public transport. Several performance indicators accompany the agreements. For instance, performance indicators comprise target patterns of urban growth and densification, citizens'

behaviours (for instance share of journeys by public transport), and $CO_2$ emissions in particular locations. The performance indicators play an important role in financing mechanisms: If important targets are not met, the municipality may have to reimburse the subsidised investments. In parallel, municipalities are also committed to accept toll roads in order to collect revenue for the transport authority (see [44] for details on managerial challenges).

The regional transport funding does not cover the full cost of new public transport facilities. The municipalities have therefore to top up. The state is responsible for the funding and constructing of (new) infrastructures (rails and transmission lines) and new stops, but the municipalities are responsible for commencing all surrounding area upgrade, like new squares, parks, pedestrian lines, bike lines, conversion of roads to urban streets, etc. Most municipalities have no extra funds earmarked for this. As will be detailed below, they are expected to extract contributions from the developers.

### 4.1.3. A Limited Application of the Compensation Right and the Reduction of Building Land

The Norwegian case study shows that apart from the need to finance transport infrastructure and urban (neighbourhood) upgrading, the implementation of the compact city policy might also require the limitation of the compensation right.

In Norway, the municipal masterplans have a 12-year perspective, and each 4th year, the new elected city council sets its mark on it by either adopting it with minor changes or requesting a full revision. In fast-growing urban areas, the city council will pay attention to the land-use part of the municipal masterplan and will discuss the impact of previous land use regulations on the objectives of the urban growth agreements (UGA). If the existing municipal masterplan designates areas for residential development in ways that will lead to sprawl, these areas have to be "downzoned" back to forest or agricultural purposes. For the municipalities, the downzoning is voluntarily and is not prescribed in the UGA. However, municipalities are using it to avoid the severe penalties they would face if the superior authorities would conclude that their planning practices and results are not in line with what is expected in the agreements.

The municipalities are entitled to downzone municipal masterplans without compensating the landowners. However, this of course creates massive pressure on city council members as no landowner lets the prospect of land value increase be removed without protests. The fact that the Norwegian planning and building law gives the municipality the authority to downzone without compensation is fundamental for the achievement of the ambitions of the UGA. In many municipalities, the funding-against-planning policy would not have been feasible or applied if the municipalities had to compensate landowners for the development possibilities due to former versions of their masterplans. Thus, at the scale of municipal masterplans, the law empowers collective property rights over private. The case is different at the lower levels of the local plans and detailed zoning plans (as a reminder: *områdeplan* and *reguleringsplan* in Norwegian). For those instruments, the compensation right applies, and the property owner is entitled to full compensation, including for development possibilities due to earlier land-use options that must be removed to meet planners' new spatial ambitions.

### 4.1.4. Financing Ambitions for a High-Quality Compact City by the Municipal Level: Negotiated Detailed Zoning Plans and Development Agreements (DAs)

We should remember that the funding made available by the UGA does not cover all the costs required to achieve the success of the compact growth policy. For municipalities which have entered into UGAs, it is therefore crucial to secure other sources of funding through land value capturing and, in particular, through development agreements (DAs). We must also remember that the DA is an instrument tying developer contribution to the granting of planning approvals. This instrument therefore falls into the category of developer obligations. Through DAs, municipalities can make compact living attractive by developing and improving amenities like new squares, parks, cross-cutting green corridors, new pedestrian lines and bike lanes. If they fail at this level, they will struggle to meet the

performance indicators related to urban growth and travel behaviours. This Norwegian version of land value capture and developer obligation can be considered an instrument of cost recovery as it aims to finance the actual cost of physical upgrading. Its rationale is not to redistribute land value to the community or to internalise negative externalities but to co-finance investments in public accessible, common amenities necessary for (a high) spatial quality.

As also mentioned above, Norwegian municipalities can develop and use three types of plans: the land use part of the municipal masterplan, the local plans, and the detailed zoning plans. The system ensures public monopoly on both, the municipal masterplans and the local plans. At the most detailed level, detailed zoning plans can be negotiated between a municipality and a private developer. Private parties may even draft detailed zoning plans and communicate them to the planning authority. The process of approving a detailed zoning plan includes repeated discussions, revisions, and negotiations between the planning authority and developers [28]. It is in the context of these discussions and negotiations that DAs are prepared. Once the detailed zoning plan is adopted, it is legally binding, and the DA is an inseparable part of the document. Thus, the DA is a steering paper that details when the developers construct the common amenities or, if this is what the parties wanted, pays the municipality for the work. Thanks to a juridical specification in the Planning and Building Law, the planning authority can specify the sequencing of the different steps within a development project. This sequencing is called the "rule of succession".

The current planning instrument of the DA was regulated in the 2006 amendment of the Norwegian Planning Act, after intense lobbying by developers. Since the 2006 regulation, four limitations have been imposed to municipal practices: no contribution for social services is allowed, and any contribution should be predictable, proportional, and relevant. To comply with the rule of proportionality, the municipality aims at fair distribution of costs between different developers on a situation-by-situation basis. To comply with the rule of predictability, most municipalities include a general passage in their municipal masterplans stating that they expect developers to contribute to co-financing the upgrading of urban common areas and common infrastructure. There is an ongoing discussion whether the masterplans should be more precise concerning the expected developers' contributions. However, to date, they remain at a very general level.

Compared with the rules of predictability and proportionality, the rule of relevance poses more difficulties to the municipalities. The rule of relevance requires that any contributions from developers must be *necessary* for *that* development project. The rule of relevance is particularly difficult to manage in urban transformation areas where there are many different landowners and developers operating in an uncoordinated way, all with their own time horizons, as is the case in most compact (re)developments [45]. Investments in public amenities are by nature investments for collective use. For example, a new public park will serve existing residents and newcomers from all developments within a certain radius. If the first developer in that area agrees to contribute their share of the cost and if the municipality decides to forward the rest and commence the construction, the rule of relevance actually *prohibits* the municipality from extracting a contribution from later developers. For instance, a contribution to a park that is already built cannot be considered a relevant condition for the not yet already built development project.

Municipal planning authorities are testing different ways to deal with the rule of relevance. B.I. Nordahl and H.J. Roald [46] divide the current municipal practices into *piecemeal* vs. *holistic* approaches. In the piecemeal approach, the planning authority subdivides the redevelopment area in small subsections and aligns the specific public amenities to each specific subsection. Within each subsection, the required common amenity will be constructed depending on the time horizons of the owners.

This contrasts to the holistic approach. In the holistic approach, the municipality take on the task of constructing all the required amenities when all developers within the larger redevelopment area have signed a legally binding statement ensuring that they will

contribute to the cost of the amenities when (or if) they decide to develop. The prioritising of what common amenities to construct first and last is up to the municipality and may be adjusted to what is most urgent for the residents' well-being. In contrast to the piecemeal approach, the sequencing of the holistic approach is based on the amenity's significance for residents' well-being and not on developers' decisions. In the holistic approach, the municipality takes a relatively large risk as the DA is usually phrased in a way that ensures that developers do not have to pay if they, for whatever reasons, have to cancel the development project. As a consequence, if some developers fail to start development, the municipality will not get their outlay covered.

Neither approach is perfect: Both require extensive calculations and iterative dialogues with market actors as no developer will sign the DA before they accept the cost calculations and their share in the funding scheme. The processes are time consuming and complex and require planning governance that is able to deploy financial, judicial, engineering, and competence in dialogues with developers.

### 4.2. Belgium: How to Improve Socioeconomic Efficiency in a Context of Weak Planning Tradition?

In Belgium, developer obligations are known as urban planning conditions (*conditions d'urbanisme* in French and *stedenbouwkundige voorwaarden* in Dutch) and urban planning charges (*charges d'urbanisme* in French and *stedenbouwkundige lasten* in Dutch). In this subsection, we analyse those instruments to deepen our analysis of the interactions between the spatial and the socioeconomic ambitions in urban land policy.

As our case study of Belgium cannot ignore the great respect for landownership and the weak planning efficiency that characterize the national planning system, we start the subsection with a contextual setting. This is followed by a focus on the 1962 Planning Act. The 1962 Planning Act was elaborated at the national level before the spatial planning competence was transferred to the regional level. This legislation can still be considered as the matrix of the current regional legislations. In the rest of the subsection, we discuss the differences between urban planning conditions and urban planning charges.

#### 4.2.1. A Context of Weak Planning Tradition and Strong Urban Sprawl

Compared with the Norwegian planning system, as well as with most other planning systems in Europe, a first characteristic of the Belgian planning system is the complete absence of prerogative at the national level. During the 1980s, the spatial planning competence was fully transferred from the national level to the regional level. In this domain, there are now three specific and independent legislative frameworks, as Belgium is a federal State composed of three regions: Dutch-speaking Flanders in the north (~6.6 million habitants on 13,625 km$^2$), French-speaking Wallonia in the south (~3.6 million habitants on 16,901 km$^2$), and Brussels in the centre (~1.2 million habitants on 161 km$^2$), which is a city-region, the federal capital and, officially, a bilingual region.

Despite the fact that responsibility for spatial planning is now devolved to the regions, we see strong similarities between planning objectives in Flanders and Wallonia. In both regions, regional authorities and regional masterplans clearly highlight the ambitions to reduce land take and to better control urban sprawl [47,48]. This ambition is consistent with the above-mentioned resource-oriented turn in planning [6].

A second characteristic of the Belgian planning system relates to the weak planning tradition. Compared for instance with the neighbouring Netherlands, Belgium is a country where the influence of spatial planning on land uses is poorly developed. This can be explained by historical reasons: the development of planning was locked in by political choices in favour of landownership. For instance, Belgium had to wait until 1962 to develop proper planning legislation at the state level. This weakness, which is inherently linked to the strength of private property rights in Belgian culture, has historical roots that go all the way back to the beginning of the 19th century [15].

Another key characteristic of the Belgian planning system is the importance of the instrument of the supra-local zoning plans (*plans de secteur* in French and *gewestplannen*

in Dutch). Regulatory zoning is at the heart of those plans, in which the residential zones that are legally available for housing developments are delimited. In both Flanders and Wallonia, supra-local zoning plans remain the most influential land use document in passive operational planning. They are indeed the key reference for evaluating the granting of development permits. Following the 1962 Planning Act, 48 supra-local zoning plans covering the entirety of Belgium were implemented during the 1960s, 1970s, and 1980s. In 1964, because of the mismanagement of planning in most municipalities, the national authority took the initiative to prepare these supra-local zoning plans. It was in 1987 that the last plan was definitely approved. This history explains why, since the eighties, the regional authorities have competence for this important instrument. This governance situation is quite specific as, in most countries, zoning plans are drawn up mainly by local authorities [9].

In many areas of Belgium, the residential zones of the supra-local zoning plans were largely oversized, which has of course allowed urban sprawl to prosper. This abundance of land legally available for housing developments is due notably to the application of the compensation right. Indeed, planners were "generous" on greenfield locations for residential development to avoid a ban on building development where the land could have been considered building land before the implementation of the zoning plan [49].

In Belgium, the application of the compensation right is not limited in time. It therefore still applies to the residential zones of the supra-local zoning plans. Several decades after the delimitation of these zones, the planning system still faces an oversupply of building land as the public authorities are not in a position to finance the compensations. This situation illustrates Hypothesis H4 that in urban land policy, the inability to increase socioeconomic efficiency tends to limit spatial efficiency. Indeed, for the Flemish as well as for the Walloon authorities, the oversupply of building land remains a major cause of the discrepancy between, on the one hand, strategic regional ambitions in favour of land take limitation and, on the other hand, what happens in practice in terms of building activity.

As it has been previously elaborated, such blockages are not as problematic in Norway. When Norwegian municipal masterplans are revised, planners can rezone—downzone—constructible land back to non-constructible land without having to consider the compensation right. This opportunity to put collective rights over private property rights is one key to the success of Norwegian planning system in the concrete implementation of the compact city paradigm, even if it has some political "costs".

### 4.2.2. The 1962 Planning Act and the Development of Urban Planning Charges

As already mentioned, Belgian planners had to wait until 1962 for the creation of proper legislation at the national scale. In 1962, the Belgian legislators decided to integrate the instrument of the urban planning charge in this legislation [50]. In both Flanders and Wallonia, the notion of urban planning charge has evolved with, respectively, the 2009 and the 2017 revisions of the regional legislations. In the remainder of the text, we will use the UPC1 abbreviation to refer to the initial instrument, and we will use the UPC2 abbreviation to refer to the current ones. In parallel, we will use the UPCo abbreviation to refer to urban planning conditions.

As the Norwegian development agreement, UPC1 can be considered developer obligations. Following the 1962 legislation, a UPC1 could be imposed on a developer in return of a development permit. One finds here a logic of cost recovery related to development-control decisions. In the context of the 1962 legislation, this instrument was justified by problems of road servicing, with the fact that urban and property developments were not always sufficiently equipped. Such situations were denounced by the preparatory parliamentary activities of the 1962 Planning Act, where it was mentioned that, "*due to inadequate land servicing, the municipal authorities are forced to carry out expensive work at the community's expense in order to remedy those problematic situations*" (own translation of the parliamentary activities).

Consideration of the justifications behind the UPC1 instrument shows an intermeshing between spatial and socioeconomic ambitions. On the one hand, the main motivation was to achieve spatial quality with proper road facilities. When the legislation was prepared, in the 1950s and early 1960s, the spatial ambitions were rather limited to an accessible roadway with electricity and water distribution. So truly we are far from being on the same level as the high-quality compact city ambitions discussed in the Norwegian case. In parallel, the parliamentary activities show that it was not considered legitimate to achieve those spatial ambitions only through public budgets. This consideration is somewhat related to the objective of socioeconomic efficiency, although it arose from public budget concerns rather than from deep philosophical reflections on the issues of property right and unearned increment. We are here therefore in the presence of an indirect rather than a direct value capture instrument [50].

### 4.2.3. The Differentiation between Urban Planning Condition (UPCo) and Urban Planning Charge (UPC2)

In 2009, the revision of the Flemish Planning Act led to the explicit distinction between the two instruments of the urban planning condition (UPCo) and the urban planning charge (UPC2). In 2017, a similar explicit distinction was integrated into the Walloon Planning Act. The analysis of the current legislation shows that in both regions, UPCo can be assimilated to the initial notion of charge (UPC1) dating from the 1962 legislation. We find here developer obligations that relate to spatial ambitions in terms of proper road equipment (water, electricity and solid pavement). In practice, the main difference between the 1962 context and the contemporary context relates to the importance of wastewater treatment infrastructure, in conjunction with the increasing importance of environmental concerns and ambitions in land-use planning. More generally, in terms of spatial quality, the difference between the current situation and the 1962 situation illustrates that planning (pre)conditions to grant a development permit depend greatly on cultural norms that vary in both time and space.

Concerning the UPC2, the current Flemish legislation mentions: "The administrative authority that delivers the permit can link charges [UPC2] to the permit. These charges [UPC2] arise from the benefit that the applicant can draw from the permit and from the supplementary tasks that the development of the project brings to the authorities" (own translation, art. 112). In the Walloon legislation, the article D.IV.53 is dedicated to the current notion of UPC2. It starts with the following paragraphs:

*"In addition to the conditions necessary for the feasibility or integration of the project, the competent authority may subordinate the issuance of the development permit to the charges* [UPC2] *it deems useful to impose on the applicant in accordance with the principle of proportionality.*

*The urban planning charges* [UPC2] *consist of acts or works imposed on the applicant, to the exclusion of any cash contribution, in order to compensate for the impact that the project imposes on the community at the municipal level"* (own translation).

Analysing the current regional legislations leads to three main observations. Firstly, we see that Walloon and Flemish authorities have both considered it necessary to develop a new tool in addition to UPC1 (or UPCo since the 2009 and 2017 reforms). This situation can be explained by the fact that the Belgian municipalities tended to impose excessive requirements, with an increasingly diverse range of developer obligations that lay outside the scope of the 1962 legislation. While basic road servicing has remained the norm for decades, some municipalities became increasingly demanding at the turn of the millennium. This observation is in line with the general observation of R. Alterman that indirect value capture policies tend to expand and intensify. Concerning Belgium, it cannot be explained by a significant development of new spatial ambitions, as exemplified by the Norwegian case. In contrast, it can be explained by the conjunction of increasing public budget constraints and increasing property prices. The rise in property values, which in Belgium was particularly strong between 1998 and 2007, was clearly higher than the inflation in

construction cost. This has created new margins in land value and some municipalities have sought to indirectly benefit from this.

The second observation is that the juridical justification behind the UPC2 instrument relates to the socioeconomic dimension rather than to the spatial dimension of land policies. On this subject, the UPC2 perspective diverges from the perspective of UPC1 and UPCo, as well as from the perspective of the Norwegian development agreements (DAs). Indeed, rather than to achieve specific spatial ambitions, the main rationale behind UPC2 is to internalise externalities. In Wallonia, the key justification is to "*compensate for the impact that the project imposes on the community*". In Flanders, the justification is based on the fact that the applicant can draw benefits from the permit but also because he or she has to contribute to the "*supplementary tasks*" that his or her project imposes on the public authorities.

The third observation, still about UPC2, is that the justifications behind it confirm the other general conclusion by R. Alterman [25] (p. 777) that "*real-life application of indirect instruments often contains ambiguities*". In the present cases, the ambiguities affect both the operationalisation and the efficiency of the instrument. For instance, in Wallonia, since the existence of the new legislation in 2017, the regional authorities have tried to operationalise the impact compensation perspective. However, this task is not only technically but also politically highly complex and it is not yet at an end stage. As a consequence, the municipalities continue to use UPC2 without clear guidelines from the regional supervision.

In the Walloon case, the task of calculating "*the impact that the project imposes on the community at the municipal level*" is made dramatically more difficult by the fact that the construction and property industry has influenced the legislators to introduce the positive externalities into the calculations. Indeed, article D.IV.53 also mentions that: "*The project's positive impacts on the community, namely its contribution to meeting a need for public interest, are taken into account in order, if necessary, to counterbalance the negative impacts*" (own translation). To our knowledge, the Walloon case is unique in the fact that the monetary evaluation of the positive externalities can be subtracted from what is imposed on the developers.

Despite this orientation towards the socioeconomic ambitions in land policies, the analysis of existing practices by Walloon municipalities shows that an efficient use of UPC2 requires the development of clear spatial ambitions through municipal masterplans. Due notably to the impossibility of receiving cash contributions (see above the article D.IV.53), the municipalities must elaborate clear guidelines on their spatial development to make the best of the potential investments UPC2 offers. In our case, we must note that most Walloon municipalities have not (yet) developed appropriate strategic plans to clearly guide the investments. However, this situation shows that increasing socioeconomic ambitions might lead to more precise masterplans and thus, potentially, to an increase in spatial ambitions. We see here a possible confirmation of Hypothesis H2: in urban land policy, increasing socioeconomic efficiency tends to increase spatial efficiency. Additionally, the situation described also illustrates Hypothesis 3: in urban land policy, the inability to increase spatial efficiency tends to limit socioeconomic efficiency. Indeed, the municipalities that are not equipped to develop appropriate strategic plans will not be equipped to significantly benefit from UPC2.

## 5. Discussion

This discussion section starts by comparing the Norwegian and the Belgian developer obligations. In the following subsection, we evaluate the Norwegian and the Belgian planning systems with respect to spatial efficiency and socioeconomic efficiency. The conceptualised ideal planning system serves as a benchmark for this evaluation.

### 5.1. Comparison between the Norwegian and the Belgian Developer Obligations

Table 1 synthesises the key characteristics of the Norwegian and the Belgian developer obligations. The central objective of the Norwegian development agreements (DAs) is closer to the Belgian UPC1 and UPCo than to the Belgian UPC2. It is based on spatial quality (pre)conditions for the development rather than on other forms of rationales that

are hard to operationalise, such as "impacts" or "externalities". Despite this resemblance in justification, the analyses show strong divergences in their concrete applications when DAs are compared with UPC1 and UPCo. A first divergence relates to the level of spatial ambitions. As DAs are applied in order to materialise the model of the qualitative compact city, the associated level of spatial ambitions is particularly high.

**Table 1.** Comparison between the Norwegian and the Belgian developer obligations.

| | Spatial Ambitions in Land-Use Planning | | Socioeconomic Efficiency in Public Finance Policies |
|---|---|---|---|
| Development agreement (DA) in Norway | - <br> - <br><br> - | The main objective relates to spatial ambitions <br> Applied in a context of high spatial ambitions related to the compact city paradigm <br> Negotiated in the plan-making process (pro-active approach) | - Socioeconomic efficiency insured due to the high spatial ambitions of the compact city paradigm |
| Urban planning charge (UPC1) and urban planning condition (UPCo) in Belgium | - <br> - <br><br> - | The main objective relates to spatial ambitions <br> Applied in a context of limited spatial ambitions (limited to proper road facilities) <br> Negotiated for the granting of the development permit (passive approach) | - Socioeconomic efficiency limited due to the limited spatial ambitions |
| Urban planning charge (UPC2) in Belgium | - <br><br> - | The need to increase socioeconomic efficiency tends to reinforce spatial ambitions <br> Negotiated for the granting of the development permit (passive approach) | - The main objective relates to socioeconomic efficiency <br> - Ambiguous legal justifications |

We must remember that the Norwegian municipalities can use three types of plans: the masterplans, the local plans, and the detailed zoning plans. The detailed zoning plans are developed at the neighbourhood scale, and they are the most detailed plans in the plan hierarchy. This type of plan gives classical prescriptions related to land uses such as the functions of the buildings, their heights, or their aesthetics. In addition to these classical regulations, the detailed zoning plans also discuss DAs in detail. While DAs are discussed in the plan-making process, UPCo as well as UPC2 are discussed at the later stage of the granting of the development permit. This difference in the phasing of the planning and development processes indicates that UPCo and UPC2 are less structurally integrated into planning practices than DAs. In fact, as the local authorities in Belgium are mainly reacting to the applications of the developers, UPCo and UPC2 must be considered as passive instruments. In contrast, local authorities in Norway play a more pro-active role, as they prepare DAs earlier, when the planning process is still in the plan-making stage and not yet at the analysis of the concrete applications. This proactive dimension is also illustrated by the fact that DAs can be discussed much earlier, during the preparation of the municipal masterplans.

The scale of urban design represents another difference between DAs and the Belgian instruments. Concerning DAs, we can recall the differentiation between the piecemeal approach and the holistic approach. With the piecemeal approach, the scale of urban design is similar to the scale of urban design for UPCo and UPC2. In contrast, in the case of the holistic approach, urban design is developed on a wider territory. By considering a wider territory, the adjustment of developer obligations to the needs of the resident population is more effective, and urban planning can therefore be considered more ambitious. However, with the holistic approach, the downside is that project negotiations with the developers are riskier for the planning authority.

*5.2. Evaluation of the Norwegian and Belgian Planning Systems with Respect to Spatial Efficiency and Socioeconomic Efficiency*

Table 2 highlights the main similarities and differences between the two case studies. This table shows that, despite strong similarities between the two analysed countries, the Norwegian planning system has more similarities with the conceptualised ideal planning system than the Belgian planning system. In particular, what we see in Norway is that high spatial ambitions go hand-in-hand with the need to ensure socioeconomic efficiency (Hypothesis H1) (see Section 5.1).

**Table 2.** Similarities and differences between the two analysed planning systems.

| Norway | Belgium |
|---|---|
| *Similarities* | |
| - Great respect for individual property rights<br>- Limited proactive public land policies and a market-oriented land-use planning<br>- It is typically considered by most Norwegian and Belgian stakeholders that public authorities should not capture the unearned increment in land value<br>- At the strategic level of planning, both countries have the spatial ambition to limit urban sprawl | |
| *Differences* | |
| - The planning system has been able to limit the consumption of the land resource and to favour an evolution towards the model of the compact city. | - The planning system has failed to curb urban sprawl or, more generally, to implement spatial ambitions and concretely influence land uses |
| Socioeconomic efficiency in the planning system is insured by the high spatial ambitions of the model of the compact city (Hypothesis 1) (see Section 5.1) | Socioeconomic efficiency in the planning system is limited due to the limited spatial ambitions (Hypothesis 3) (see Section 5.1) |
| - The UGA illustrates the ability of the Norwegian planning system to influence other public policies such as transportation and public finance | - The Belgian planning system usually fails to influence other public policies. As a consequence, important sector-based policies, in the domains of transportation, housing and taxation, continue to feed urban sprawl |
| - Implementing UGA at the national level of the planning system has resulted in an increased attention of the local level in favour of the model of the compact city. This has leaded to both, an increased attention to developer obligations and a reduction of the building land supply | - A strict application of the compensation right limits the possibility to reduce an overabundant supply of building land. This is illustrative of Hypothesis 4 that the inability of a planning system to increase its socioeconomic efficiency tends to limit its spatial efficiency |

Although the high spatial ambitions inherent to the compact city paradigm have significant impacts on the socioeconomic efficiency in Norwegian urban land policy, we must state that we are unable to actually measure this efficiency. Indeed, as most negotiations between developers and municipalities remain opaque, we are unable to give reasonable estimations on how development agreements actually contribute to land value capturing. On the basis of the available informal information, it can be considered that the compact city ambitions put pressures on the developers' margins. However, nothing indicates that, in most property development, what is imposed to the developer lead to redistribute all land value to the community ... as it should be in an ideal planning system.

In Norway, the state impetus in favour of the compact city model is reflected in terms of strategic planning at the national and regional levels, but it is above all reflected by the ability of the planning system to influence other public policies such as transportation and public finance. In the major metropolitan areas of the country, this is clearly illustrated by the implementation of a powerful funding-against-planning instrument such as the urban growth agreement (UGA). Nothing similar can be found in Belgium, where important sector-based policies in the domains of transportation, housing and taxation continue

to feed urban sprawl [33,48]. From that standpoint, the comparison between the two countries shows that the Norwegian planning system is the closest to the conceptualised ideal planning system, despite inevitable needs of improvement.

Implementing UGAs has resulted in an increased focus of the Norwegian municipalities on developer obligations. With this increased attention, we observe a virtuous cycle through a financial lever where the resources made available through the development agreements with the developers are helpful for securing the resources made available through the UGA with the regional transport authorities. This observation relates to the interactions between the spatial and the socioeconomic ambitions in urban land policy. In particular, it confirms our initial assumption that there are processes of mutual strengthening between the two dimensions.

In Norway, the implementation of UGA has resulted in an increased attention to developer obligations, but it has also impacted building limitation. Under the threat of UGA, municipalities prove to be more likely to downzone greenfield building land back to non-constructible land. Again, nothing similar can be found in Belgium, where a strict application of the compensation right limits the possibility to reduce an overabundant supply of building land. This situation illustrates hypothesis 4 that in urban land policy, the inability to increase socioeconomic efficiency tends to limit spatial efficiency. In terms of public finance, such situation also depicts a particularly inefficient planning system as compensations are due although value capture has not previously taken place.

In Belgium, we also see forms of positive interactions between the spatial and the socioeconomic ambitions in urban land policy. Compared with Norwegian municipalities, the motivations of the Belgian municipalities appear more socioeconomic than spatial. Even still, our investigations on the Belgian context have also confirmed our assumption of mutual strengthening between spatial efficiency and socioeconomic efficiency. In particular, we observe that the wishes of the municipalities to obtain greater profits from developer obligations might lead to increased spatial ambitions through the necessity to elaborate more integrated and sophisticated masterplans. This situation actually relates to the Hypothesis 2.

## 6. Conclusions

This article has addressed the issue of the interactions between spatial efficiency and socioeconomic efficiency in urban land policy. Our exploratory research started with the observation that to date, land policy research tends to analyse these two dimensions in an insufficiently integrated way. Our discussion confirms this diagnosis and indicates that researchers working on value capture instruments should pay specific attention to the interactions between the spatial and the socioeconomic ambitions in land policy. We consider that this recommendation can help in the development of the growing research field dedicated to land value capture [8].

Our exploratory research also started with the initial assumption that there are processes of mutual strengthening and blockages between spatial efficiency and socioeconomic efficiency in urban land policy. This initial assumption was related to four hypotheses. In order to develop a robust methodological approach to test those hypotheses, we developed a theoretical framework with respect to spatial efficiency and socioeconomic efficiency in land policy. This framework depicts an ideal planning system where land policies efficiently contribute to the achievement of the land uses spatial planners seek and, simultaneously, efficiently contribute to redistributing the unearned increments to the community. This ideal planning system has served as a benchmark for an exploratory comparative approach where Norwegian and Belgian developer obligations were compared. At the end of this exploratory examination, we consider that this reference model can serve as a benchmark for further analyses on various planning systems.

The comparison between the two analysed countries shows that, compared with Belgum, Norway has more similarities with the conceptualised ideal planning system. In Norway, the planning policy achieves better results in view of the resource-based turn

thanks to clear and strong political impetus from the national level. This impetus in favour of the compact city model has resulted in municipalities' increased attention to land value capture through developer obligations and, therefore, to an increase in the capacity of local governments to redistribute lands' value to their communities.

It can also be seen from the two case studies that our initial assumption of mutual strengthening between spatial efficiency and socioeconomic efficiency in urban land policy is globally confirmed. In Norway, we see that high spatial ambitions go hand in hand with the need to ensure socioeconomic efficiency (confirmation of Hypothesis H1). In Belgium, we observe that the wishes of the municipalities to extract greater financial profits from developer obligations lead to increased spatial ambitions through the need to elaborate more integrated masterplans (confirmation of Hypothesis H2). The Belgian case also illustrates a confirmation of Hypothesis H3, as the municipalities unable to increase spatial efficiency through the development of appropriate strategic plans are also unable, in socioeconomic terms, to make the best of the value capture instruments at their disposal. With the Belgian case, it can also be seen that there are blockages between socioeconomic (in)efficiency and spatial (in)efficiency (confirmation of Hypothesis H4). In this country, as opposed to Norway, a strict application of the compensation right limits the possibility of reducing the overabundance of building land. For decades, this blockage has hindered the effective control of urban sprawl.

If the results of the two case studies indicate that our hypotheses are generally confirmed, further investigations are of course needed to deepen our exploratory discussion on the topic. From this perspective, the availability of robust quantitative data on developer obligations could be greatly helpful for strengthening the proposed conceptual and methodological approach. Indeed, such quantitative data should allow for giving quantitative estimations of the proportion of the land value that is actually redistributable to a community or, in other words, of the socioeconomic efficiency in land policy.

We should also remember that in a context of strong proactive planning, Dutch research has brought to light the threat of the "double hat". This threat is that public authorities might put emphasis on financial ambitions at the expense of land-use planning ambitions (this goes against Hypothesis H2). This observation contrasts with our initial assumption, and it therefore illustrates the need for further research aiming to clarify the complex interactions between the spatial and the socioeconomic ambitions in urban land policy.

**Author Contributions:** Formal analysis, J.-M.H., B.I.N. and M.B.H.; Methodology, J.-M.H.; Writing—original draft, J.-M.H. and B.I.N.; Writing—review & editing, J.-M.H. All authors have read and agreed to the published version of the manuscript.

**Funding:** This research received no external funding.

**Institutional Review Board Statement:** Not applicable.

**Acknowledgments:** This article is based on work from COST Action PuVaCa (CA17125), supported by COST (European Cooperation in Science and Technology), www.cost.eu, accessed on 19 September 2022.

**Conflicts of Interest:** The authors declare no conflict of interest.

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
