# Peer review of "Spatial Efficiency and Socioeconomic Efficiency in Urban Land Policy and Value Capturing: Two Sides of the Same Coin?"

_sustainability, doi:10.3390/su142113987_

Round 1

Reviewer 1 Report (Previous Reviewer 1)

Dear authors, thank you for allowing the correction of your article.

The introduction is correct, and the hypotheses put forward are well constructed.

State of the art lacks more depth in terms of academic references. It would be interesting to know other approaches to the measurement of socioeconomic efficiency and spatial efficiency to know the methods and the different theoretical debates that exist.

The methodology suffers from several problems. The method of analysis to be followed is not stated. In the case of using a qualitative approach, it would have been appreciated if the elements of research and how they are validated were established. There is very interesting software for these tasks. As far as efficiency measurement is concerned, qualitative methods are very valid. I suppose that some panel must have been built where the different categories are collected to be analyzed. 

The study results are descriptive without establishing elements that investigate how they influence the hypotheses to be validated.

It would have been interesting to include a map with each country's location and a table with the basic socioeconomic data of each one.

I have a doubt, what are the legislative and executive execution differences in each state analyzed? Do they influence the results of the research?

The construction of analysis tables with categories is advantageous; a timid attempt has been made in the discussion. In fact, in my opinion, this is not such a discussion. It is part of the results. The discussion should emphasize the results with the different theoretical approaches.

The results are correct based on a descriptive study. Still, as I have already said, I believe that the research needs to provide solid data supported by a more focused and validated methodology.

Regards

Author Response

See the enclosed file.

Reviewer 2 Report (Previous Reviewer 2)

The authors still did not specify the evaluation criteria, as I have already stated, I would recommend supplementing them.

In the end, the authors did not accept the specification of the term "our working hypothesis", I still recommend either specifying the term or stating only "our established hypotheses").

Author Response

See the enclosed file.

Reviewer 3 Report (Previous Reviewer 3)

Dear Authors,

Please reconsider writting the list of references.

At p.5 there is a writting mistake in the paragraph "buildings” [21] (p.

E.1).", which page number is?

The manuscript has been improved, well done.

Author Response

See the enclosed file

Round 2

Reviewer 1 Report (Previous Reviewer 1)

Dear authors, thank you for your answers. 

First of all, I value the epistemological debate around methodology. Although I believe that the study could have been approached differently, showing quantitative conclusions, I understand the validity of your research.

I still believe that the study is descriptive and that the differences and similarities of the cases studied should be highlighted using tables (such as the one included in the discussion). Visualizing the results is essential; for easier reading.

The conclusions are adequate, but the limitations of the study and its extension/usefulness to other cases should be emphasized, as well as the way to approach this problem in the case of a similar study.

Good luck

Author Response

See the enclosed file.

This manuscript is a resubmission of an earlier submission. The following is a list of the peer review reports and author responses from that submission.

Round 1

Reviewer 1 Report

The abstract has gained in concreteness although, in my opinion, it should be a little shorter.

The number of keywords is too high, define them more precisely. They should not exceed six.

The new introduction draws an interesting picture and focuses the research clearly and directly.
Note that the citations are not in the correct format.

The literature review has undergone important modifications that are to be welcomed in point 2.2.1. In your first paragraph (line 189 I think you should include a citation to support this statement).
The same should be done with the statements in lines 221-225 and 225-228. The value of a good review is to establish the framework in which research has been and is being developed concomitantly with that which has been carried out. The development of the text has been improved but more references are needed. Note for example the final paragraph of point 2.2.1 that does not make any reference when it is essential to do so.

The methodology is now clearer and more direct. I am concerned that a method has not been used to obtain a more obvious quantification of the results without abandoning the perspective used.  Including a map would be interesting.

The results are very descriptive, the problem is that the comparative summary table included in the discussion should appear in the results section as it is self-descriptive.

The discussion is interesting, it focuses on the comparison of both models, but should include the academic debate on the approach taken as well as criticisms of both systems. As it is currently presented, it is not part of the conclusions.

The conclusions are correct, although too much emphasis is placed on why the case studies have been selected when this has already been adequately justified. It is more interesting to point out the limitations that two such different territories can produce in the study.
The last paragraph should be moved since what is interesting is to start talking about the hypotheses established in the study.

Possible future lines of study should be included as well as the limitations that a study of this type may entail.

Good luck

Reviewer 2 Report

Hypothesis 1 and 3, resp. 2 and 4 are identical, only expressed in a different way (positive and negative evaluation of the same factor)
There is no understandable concept, it would be appropriate to add in the article why the hypotheses were stated. Furthermore, it would be appropriate to specify the procedure by which the authors want to confirm the hypotheses, resp. refute.
The paper lacks the criteria by which the authors want to refute or confirm the hypotheses.

In conclusion, a specific evaluation of the hypotheses is missing.

The authors talk about the working hypothesis, which they evaluate in the end, but it is not clear which hypothesis it is.  (If the results of the two case studies indicate that our working hypothesis are gener-ally confirmed, further investigations are of course needed to deepen our exploratory discussion on the topic.)
The authors use a large number of self-citations.

Reviewer 3 Report

Six keywords are enough;

The Conclusion section should be improved by adding previous research work and comparing with the current findings;

Reviewer 4 Report

The list of references contains some minor typographical mistakes.